# Mechanical Properties and Deformation Mechanisms of Graphene Foams with Bi-Modal Sheet Thickness by Coarse-Grained Molecular Dynamics Simulations

**DOI:** 10.3390/ma14195622

**Published:** 2021-09-27

**Authors:** Shenggui Liu, Mindong Lyu, Chao Wang

**Affiliations:** 1School of Mechanics and Civil Engineering, China University of Mining and Technology, Beijing 100083, China; liushg2002@163.com (S.L.); zeqiulmd@126.com (M.L.); 2LNM, Institute of Mechanics, Chinese Academy of Sciences, Beijing 100190, China; 3School of Engineering Science, University of Chinese Academy of Sciences, Beijing 100049, China

**Keywords:** graphene foam materials, microstructure, bi-modal sheet thickness, stress-strain curve, coarse-grained molecular dynamics

## Abstract

Graphene foams (GrFs) have been widely used as structural and/or functional materials in many practical applications. They are always assembled by thin and thick graphene sheets with multiple thicknesses; however, the effect of this basic structural feature has been poorly understood by existing theoretical models. Here, we propose a coarse-grained bi-modal GrF model composed of a mixture of 1-layer flexible and 8-layer stiff sheets to study the mechanical properties and deformation mechanisms based on the mesoscopic model of graphene sheets (Model. Simul. Mater. Sci. Eng. 2011, 19, 54003). It is found that the modulus increases almost linearly with an increased proportion of 8-layer sheets, which is well explained by the mixture rule; the strength decreases first and reaches the minimum value at a critical proportion of stiff sheets ~30%, which is well explained by the analysis of structural connectivity and deformation energy of bi-modal GrFs. Furthermore, high-stress regions are mainly dispersed in thick sheets, while large-strain areas mainly locate in thin ones. Both of them have a highly uneven distribution in GrFs due to the intrinsic heterogeneity in both structures and the mechanical properties of sheets. Moreover, the elastic recovery ability of GrFs can be enhanced by adding more thick sheets. These results should be helpful for us to understand and further guide the design of advanced GrF-based materials.

## 1. Introduction

Graphene is the thinnest-known sheet, composed of a single layer of carbon atoms, which has been used as nanoscale building blocks to fabricate a good deal of macroscale bulk materials, such as one-dimensional graphene fibers [1,2,3,4], two-dimensional graphene papers [5] and three-dimensional (3D) graphene foams (GrFs) [6,7,8,9,10,11,12,13,14,15,16]. Among them, 3D GrFs have attracted much attention in recent years from material scientists to engineers due to their combined properties of excellent electrochemical stability [17], superior electrical conductivity [18], high energy absorption [19] and broad application prospects in advanced materials [20], stretchable electronic devices [21] and energy-storage components [22].

It is shown that the mechanical and physical properties of graphene sheets, as well as their macro-assembles of GrFs are greatly influenced by thickness of sheets. For example, for a single graphene sheet, the bending stiffness is proportional to the third power of thickness if there is no interlayer slippage, according to the theory of plates [23,24]. If the effect of interlayer slippage cannot be neglected, the stiffness of multilayer graphene sheets is still highly dependent on thickness based on the experimental study [25]. Roughly speaking, the thicker the graphene sheet, the greater the stiffness. The electrical conductivity of a single graphene sheet increases with an increased thickness according to the first principle calculation [26]. For their macro-assemblies of GrFs, Wang et al. [27,28] found that both elastic properties and Poisson’s ratio of GrFs increased with an increased thickness of sheets using coarse-grained molecular dynamics simulations (CGMD). Chen et al. [21] experimentally found that the thickness-dependent conductivity of GrFs reached the maximum value of ~10 S/cm at an optimal number of graphene layers of around 5, which was reproduced later by Liu et al. [26] using a combination of the first principle calculations and CGMD simulations.

Due to complicated preparation processes involving redox reactions and the growth/aggregation of graphene sheets by chemical vapor deposition (CVD) [10,29] or the self-assembly method [8,30,31], GrFs are always composed of sheets having multiple thicknesses rather than a single thickness. For example, Nieto et al. [10] made a freestanding GrF consisting of graphene sheets with ~3–10 layers, which was grown onto a nickel foam structure by a CVD method. Zhao et al. [32] made a GrF composed of sheets with 1–4 graphene layers by CVD and electrochemical strategies. More experimental reports about varied thickness of graphene sheets in GrFs can be found in [8,21,30,33,34].

However, to our best knowledge, all numerical models of GrFs published in references, no matter the full-atomic models [16,35,36,37,38], coarse-grained models [28,39,40,41,42] or finite element one [43], have a fairly idealized assumption that the thickness of constituent graphene sheets is the same throughout GrF systems. For example, all samples using full-atomic models [16,35,36,37,38] are composed of thinnest 1-layer graphene sheets, while other coarse-grained models [28,39,40,41,42] or finite element one [43] are composed of sheets with 1-layer or multi-layer single thickness. Although the GrFs composed of graphene sheets with multiple thicknesses have been widely observed in practical systems as mentioned above, the mechanics of this kind of GrFs have not been studied up to now, and the role of graphene sheets of different thicknesses in the system, as well as the synergy between them, remains elusive.

In this paper, a so-called bi-modal GrF model composed of 1-layer flexible sheets and 8-layer stiff sheets are proposed to systematically study these essential issues based on the coarse-grained model of graphene sheets with varied thickness proposed by Cranford et al. [44], as given in Section 2. Section 3 gives the tensile stress–strain curves of GrFs with varied proportions of stiff sheets, and the modulus increases almost linearly with an increased proportion of stiff sheets; there exists a critical proportion of stiff sheets ~30%, at which the strength reaches the minimum value. This critical phenomenon is explained by the analysis of structural connectivity by stiff sheets and the deformation energy of bi-modal GrFs. Furthermore, we find that high-stress states always emerge in stiff sheets, while flexible sheets experience larger deformation. The conclusion is given at the end.

## 2. Numerical Model of Bi-Modal GrFs and Methodology

We adopted the coarse-grained graphene model developed by Cranford et al. [44] and proven effective and computationally efficient in a series of studies including the mechanical deformations of a single graphene nanoribbon [44] and graphene macro-assemblies [28,45,46,47]. In this scheme, a square graphene sheet with the side length of 2.5 nm containing 264 carbon atoms Figure 1a-i is reduced to be a so-called coarse grain. So, a larger square graphene sheet with a side length of 75 nm can be represented by only 900 coarse grains connected by a set of linear springs and angle ones as shown in Figure 1a-ii,a-iii. The total energy of a graphene sheet is calculated by $E_{\mathrm{total}}=\sum_{i=1}^{N_{B}}E_{B}^{i}+\sum_{i=1}^{N_{\varphi }}E_{\varphi }^{i}+\sum_{i=1}^{N_{\theta }}E_{\theta }^{i}+\sum_{i=1}^{N_{\mathrm{LJ}}}E_{\mathrm{LJ}}^{i}$, where *N*_B_, *N*_φ_, *N*_θ_ and *N*_LJ_ are the number of bonds (linear springs), in-plane angles, out-of-plane angles and bead pairs, respectively; $E_{B}^{i}$, $E_{\varphi }^{i}$, $E_{\theta }^{i}$ and $E_{\mathrm{LJ}}^{i}$ are the bond energy, in-plane angle energy, out-of-plane angle energy and van der Waals energy, respectively, and are calculated by Equations (1)–(4) in simulations; the parameters of $k_{B}$, $k_{\varphi }$ and $k_{\theta }$ are the stiffness of the linear spring for bond stretching, the angle spring for in-plane shearing and the out-of-plane bending of sheet, respectively; the equilibrium distance of two bonded beads $r_{0}$ is 25 Å; the in-plane equilibrium angle $\varphi_{0}$ between two bonds connecting three neighbor beads is 90°; the out-of-plane equilibrium angle $\theta_{0}$ between two bonds connecting three neighbor beads is 180°; ε is the depth of the potential well, and σ represents the zero-energy distance between two beads in the van der Waals potential function. *r* is the current distance between two beads. In practical GrF materials, a constituent graphene sheet contains about 1–10 graphene layers [10,21,48]. Here, a GrF composed of a mixture of 1-layer flexible sheets and 8-layer stiff sheets is used to study the mechanics of bi-modal GrFs. All mechanical parameters for the coarse-grained 1-layer and 8-layer sheets in our simulations are listed in Table 1, which was obtained based on the equivalent energy principle by Cranford et al. [44].
(1)$$E_{B}{=k}_{B}{\left(r-r_{0}\right)}^{2}/2$$
(2)$$E_{\varphi }{=k}_{\varphi }{\left(\varphi -\varphi_{0}\right)}^{2}/2$$
(3)$$E_{\theta }{=k}_{\theta }{\left(\theta -\theta_{0}\right)}^{2}/2$$
(4)$$E_{\mathrm{LJ}}=4\varepsilon ({\left(\sigma /r\right)}^{12}-{\left(\sigma /r\right)}^{6})$$

Using the numerical synthesizing method described in detail at the end (Appendix B: Preparation of Numerical Model) of this paper, we obtain a series of well-equilibrated GrFs with a tunable proportion of 8-layer sheets, i.e., *v* = 0, 30%, 50%, 70% and 100%. Figure 1b shows the initial state of the GrF with the proportion of 8-layer sheet *v* = 50%, in which graphene sheets are randomly distributed in the foam system similar to that observed in a SEM experiment [10]. In order to mimic the strong connection between neighbor sheets due to strong physical crosslinks or chemical functional groups [10,49] in practical GrF systems, we adopt a crosslink model which has been widely used to study the large-deformation behavior and fracture mode of both carbon nanotube buckypapers [49,50] and graphene foams [27,46]. As shown in Figure 1b, a certain amount of crosslinks (green color) characterized by Equation (5) are added between neighbor sheets in all GrF samples to enhance inter-sheet connections. $k_{C}$ is the spring constant, and *r* is the current distance between two beads in different flakes with a equilibrium distance $r_{0}$ = 25 Å. Furthermore, we assume that the bonds in sheets as well as crosslinks between neighbor sheets would break when the local tensile strain surpasses the critical value of 12% according to the experimental and theoretical researches [51,52]. In addition, the equilibrium density of all GrF systems ranges from 46 to 303 mg/cm^3^, which is comparable to that obtained in experimental systems in the range of 1–1400 mg/cm^3^ [6,38,48].
(5)$$E_{C}{=k}_{C}{\left(r-r_{0}\right)}^{2}/2$$

The four typical microstructures in bi-modal GrFs, i.e., edge–edge (E–E), edge–surface (E–S), point–surface (P–S) and surface–surface (S–S) for both flexible sheets and stiff sheets are shown in Figure 1c. The proportions of S–S contacts with a larger contact area slightly decrease while the proportions of E–S and E–E and P–S contacts with a smaller contact area slightly increase as an increased fraction of stiff sheets in GrFs as shown in Figure 1d. Accordingly, the areal density (the contact area divided by the system volume) decreases with the increased proportion of stiff sheets.

All systems are equilibrated by energy minimization and then relaxation at 300 K and 1 atmosphere under an isothermal–isobaric (NPT) ensemble with the criterion that the total energy fluctuation converges to less than 1%. In equilibration, the periodic boundary conditions are imposed in all three directions of the simulated samples. In the loading process, samples are uniaxially tensioned in the x-axis direction with a tensile strain rate 4.3 × 10^6^ $s^{-1}$ with a zero-pressure barostat in the other two unloading directions at room temperature. A time step of 10 fs is used. All simulations are performed in the large-scale atomic/molecular massively parallel simulator (LAMMPS) [53], and the results are visualized based on the Open Visualization Tool (OVITO) [54].

## 3. Results

### 3.1. Stress–Strain Curves in Uniaxial Tension

Figure 2a shows the uniaxial tensile stress–strain curves of the GrFs with varied proportions of 8-layer stiff sheets *v* = 0, 30%, 50%, 70% and 100%. The stress–strain curve of the GrF composed entirely of 1-layer sheets, i.e., *v* = 0, clearly shows four stages, as signified in the black curve: the linearly elastic stage, yielding stage, obvious hardening stage and the final fracture stage. As shown in Figure 2b-i, the snapshots in the front viewpoint at the tensile strain of 0, 0.25, 0.5 and 0.75 show that the system contracts obviously perpendicular to the tensile direction due to Poisson’s effect and breaks locally at the larger tensile strain of 0.5 and 0.75 as signified by yellow circles. For the GrFs composed of the mixture of 1-layer sheets and 8-layer ones at varied proportion *v* = 30%, 50% and 70% of 8-layer sheets, the stress generally increases with an increased *v*. Choosing the system with *v* = 50% as an example, in a wide range of tensile strain, the tensile stress is higher than that of the system containing a small amount of stiff sheets with *v* = 30%; the snapshots of the system at the tensile strain of 0, 0.25, 0.5 and 0.75 in Figure 2b-ii show that the system shrinks very little perpendicular to the tensile direction compared to the GrF composed entirely of 1-layer sheets. For the GrF composed entirely of 8-layer sheets, i.e., *v* = 100%, the tensile stress is much larger than that of all systems at any tensile strain; moreover, the system breaks seriously even at a smaller tensile strain of 0.25 as highlighted by yellow circles in Figure 2b-iii, and the system shrinks very little as the GrF with *v* = 50%. Qualitatively, it is seen that the fraction of stiff sheets greatly influences the macroscopic mechanical responses of the bi-modal GrFs.

Furthermore, the quantitative dependency between the modulus/strength of GrFs and the proportion of stiff sheets is numerically obtained and shown in Figure 2c. For each *v*, we conduct uniaxial tension simulations using five numerical samples to obtain the mean and variance of the modulus and strength of the system. The modulus increases almost linearly with the increased *v* because the stretching/shearing/bending modulus of 8-layer stiff sheets is much larger than that of 1-layer flexible sheets as given in Table 1, which is qualitatively consistent with the mixture rule [55]. Interestingly, we find that there exists a critical proportion of *v*~30%, at which the strength of the bi-modal GrFs reaches the minimum value, i.e., the strength of the GrF decreases first and reaches the minimum value at the critical *v* = ~30%, then increases monotonically with an increased *v*. In order to explain this critical phenomenon, we study the connectivity of stiff sheets in GrFs and find that the connectivity transforms from 0 to 1 as *v* > 30% as shown in Figure 2d, i.e., a complete bearing path by composed of entirely stiff sheets forms at this critical proportion, and the strength of the system would be instead determined by mechanical properties of stiff sheets. Comparing the GrFs with *v* = 0 and 30%, because flexible sheets have better deformation coordination ability, the number of broken bonds in the GrF with *v* = 0 is always smaller than that in the GrF with *v* = 30% as shown in Figure 2e, which is responsible for the larger strength of the GrF with *v* = 0. In other words, a small number of stiff sheets would deteriorate the bi-modal GrF because of their poor deformation coordination ability. For the system with *v* > 30%, the deformation of the system would be dominated by stiff sheets, because of the intrinsic high strength of stiff graphene sheets, the strength of the system increases monotonically with an increased proportion *v*.

### 3.2. The Distribution and Evolution of Deformation Energy

To further understand the deformation of bi-modal GrFs, we calculate the total stretching energy $E_{B}^{\mathrm{total}}=\sum_{i=1}^{N_{B}}E_{B}^{i}+\sum_{j=1}^{N_{C}}E_{C}^{j}$, in-plane shearing energy $E_{\varphi }^{\mathrm{total}}=\sum_{i=1}^{N_{\varphi }}E_{\varphi }^{i}$, out-of-plane bending energy $E_{\theta }^{\mathrm{total}}=\sum_{i=1}^{N_{\theta }}E_{\theta }^{i}$ and van der Waals energy $E_{\mathrm{LJ}}^{\mathrm{total}}=\sum_{i=1}^{N_{\mathrm{LJ}}}E_{\mathrm{LJ}}^{i}$ for the GrFs with varied *v*, where $E_{B}^{i}$,$E_{\varphi }^{i}$, $E_{\theta }^{i}$, $E_{\mathrm{LJ}}^{i}$ and $E_{C}^{j}$ are calculated by the Equations (1)–(5), respectively. Figure 3a shows that the total in-plane shearing energy of GrFs increases slightly with an increased tensile strain and saturates at a critical tensile strain of ~0.4; in particular, it is almost independent on *v*. This is consistent with our previous finding [27] that the shearing deformation of sheets is not the main deformation mode in GrFs and that the shearing energy of GrFs always keeps a small value regardless of the characteristic of the constituent graphene sheets. For the two other kinds of deformation modes, the stretching and bending of sheets, as shown in Figure 3b,c, on the one hand, the respective elastic energy of the two modes is larger than the corresponding shearing energy in Figure 3a, which indicates that the bending and stretching of sheets are still the main deformation modes in the bi-modal GrFs, as in in the GrFs composed of a single thickness of sheets [27]; on the other hand, both kinds of energy increase with an increased *v* at any given tensile strain; here, we note that the initial crosslink density, as well as the local connectivity of neighbor sheets, are almost the same for samples, so in the early stage of tension, because there is no bond breaking, the sheets in systems would experience almost similar stretching/shearing/bending deformations due to the same constrains imposed by crosslinks; however, due to the huge difference in stiffness and strength of 1-layer flexible sheets and 8-layer stiff ones, the immediate stress level, as well as the initial modulus of the GrF with more stiff sheets (larger *v*), show larger values as observed in Figure 2a,c. Moreover, the van der Waals adhesion energy Figure 3d exhibits a similar dependency on *v* as that of the stretching/bending energy; from the structural analysis in Figure 1d, we know that the GrF with more stiff sheets is apt to be a state with less inter-sheet contact area, which is responsible for the increase of the van der Waals energy as shown in Figure 3d. For the pure GrF or the GrF with 30%, the stretching/bending energy of the system increases much slightly with an increased tensile strain, as indicated by the slope bar; in sharp contrast, for the GrF containing more 8-layer sheets, e.g., *v* = 50%, 70% and 100%, the stretching/bending energy of the system increases obviously with an increased tensile strain, especially in the stage with the tensile strain <0.4; see the respective slope bar. This difference in the change of the stretching/bending energy at the critical value of *v* = ~30% indicates that the deformation mode of the system has shifted from being dominated by flexible sheets to being dominated by stiff sheets, which is consistent with the finding by the connectivity analysis as discussed in Figure 2d.

### 3.3. The Distribution and Evolution of Local Stress and Strain

In order to understand the role of flexible sheets and stiff ones in bi-modal GrFs in tension process, we study the distribution and evolution of local stress and strain in both 1-layer flexible and 8-layer stiff sheets. Figure 4a-i shows the distribution of local von-Mises stress in both flexible sheets and stiff ones in the GrF with *v* = 50%. In the initial state with the tensile strain *ε* = 0, the system is well equilibrated; then, with an increased tensile strain from 0 to 0.8, more and more high tensile stress regions as color coded by red emerge in the system. By separately displaying the stress distribution in flexible sheets and stiff ones in Figure 4a-ii,a-iii, we find that the high-stress regions mainly locate in stiff sheets, especially at the larger tensile strain of 0.4 and 0.8, which is mainly attributed to the poor deformation coordination ability of stiff sheets compared to the flexible ones. Furthermore, we count the number of the “high-stress beads”, defined as the magnitude of local von-Mises stress σ_vm_ being larger than the average value of the system, i.e., |σ_vm_|>$\sum_{i=1}^{N}\left|\sigma_{vm}^{i}\right|/N$, where *N* is the number of beads in the system. Figure 4b shows that the fraction of high-stress beads in stiff sheets is always larger than that in flexible ones; the curve related to 1-layer sheet is not monotonous, because the bonds in 1-layer sheets are broken ceaselessly with increasing strain. It should be noted that the aggregation of high-stress regions in stiff sheets also holds in the GrFs with the proportions of *v* = 30% and 70%, as shown in Appendix A.

In order to compare the deformation behaviors of 1-layer flexible sheets and 8-layer stiff ones in the bi-modal GrFs with *v* = 50%, we calculate the average relative deformation for the three modes, bond stretching, in-plane shearing and out-of-plane bending of all 1-layer sheets and 8-layer ones by $\bar{\varepsilon_{r}}=1/N_{r}\sum_{i=1}^{N_{r}}\left|(r_{i}-r_{0})/r_{0}\right|$, $\bar{\varepsilon_{\varphi }}=1/N_{\varphi }\sum_{i=1}^{N_{\varphi }}\left|(\varphi_{i}-\varphi_{0})/\varphi_{0}\right|$ and $\bar{\varepsilon_{\theta }}=1/N_{\theta }\sum_{i=1}^{N_{\theta }}\left|(\theta_{i}-\theta_{0})/\theta_{0}\right|$, respectively, where *N*_r_, *N*_φ_ and *N*_θ_ are the number of bonds, the number of shearing angles and the number of bending angles in all flexible sheets or stiff ones. For the system with *v* = 50%, the part of flexible sheets shares the same value of *N*_r_, *N*_φ_ and *N*_θ_ with the part of stiff ones. As shown in Figure 5, the average relative deformation for the three modes of flexible sheets (black line) is always larger than that of stiff sheets (red line) in the whole tension process. For example, in Figure 5a, the mean relative stretching length of bonds in flexible sheets is in the range of 0.002–0.0035 and is larger than that in stiff sheets at any tensile strain. For a more vivid comparison, the snapshots of the flexible part and stiff one at the tensile strain of 0.4 are color-coded by the local relative deformation, as shown in the inset in each figure. It is shown that high-strain regions mainly emerge in flexible sheets; in contrast to that, high-stress regions always emerge in stiff sheets in Figure 4, which is reasonable and easily understood. We also examine the deformation behaviors of flexible sheets and stiff ones in the bi-modal GrFs with proportions of *v* = 30% and 70% and obtain the same conclusion as shown in Appendix A. Furthermore, from the snapshots in the three insets, we can see that the deformations of all sheets, especially flexible sheets, are very nonuniform, which is attributed to the high nonuniformity of foam structures and the heterogeneity of the mechanical properties of the two kinds of sheets.

### 3.4. Effect of the Proportion of Stiff Sheets on the Elastic Recovery of Bi-Modal GrFs

We also examine the elastic recovery of the GrFs with *v* = 0, 30%, 50%, 70% and 100% in the compression–uncompression process. The systems are first compressed to a maximum compressive strain of 0.8 at a loading rate of 4.3 × 10^6^ s^−1^ in x direction and then uncompressed for a long time (20 ns) to fully relax. The stress–strain curves for the five GrFs in the compression–uncompression process are shown in Figure 6a, and none of them recover completely due to the microscopic plasticity unveiled in the works [47,56]. The residual strain for the five systems is 0.74, 0.39, 0.28, 0.17 and 0.07, respectively, which decreases with an increased *v,* as shown in the inset in Figure 6a. The corresponding snapshots of the compression–uncompression process of the three GrFs with *v* = 0, 50% and 100% are given in Figure 6b,d for intuitively showing the enhancement of the elastic recovery of GrFs with 8-layer sheets. This can be explained using our previous work [27] about the GrFs composed of mono-layer graphene sheets; at a given crosslink density, more elastic energy can be stored in the GrF composed of more stiff sheets as it is compressed. So, we can increase the proportion of stiff sheets in a bi-modal GrF to obtain a good elastic recovery ability in compression in practical applications.

## 4. Conclusions

We adopt coarse-grained molecular dynamics simulation to study the mechanical properties and deformation mechanisms of GrFs with bi-modal sheet thickness distribution. The modulus of GrFs increases almost linearly with the increased proportion of the 8-layer sheets, which can be well explained by the mixture rule. The strength of the bi-modal GrF decreases first and then increases with an increased proportion of 8-layer sheets; a minimum strength is obtained at a critical proportion of *v* ~ 30%. This is because, in the case of a smaller proportion of 8-layer sheets, *v* < 30%, first, stiff sheets do not form a complete bearing path in GrF, and the strength of the system is still mainly determined by the interconnected flexible sheets; second, due to the poor deformation coordination capacity of stiff sheets, when a certain number of stiff sheets are included in the GrF, more flexible sheets break in the tension process of the system. As the proportion of stiff sheets surpasses 30%, a complete bearing path form by stiff sheets, and the strength of the system is instead determined by the properties of stiff sheets. So, the strength of the system would increase with an increased proportion of stiff sheets. We also examine the distribution and evolution of the deformation energy of the stretching, shearing and bending of sheets and van der Waals adhesion energy between neighbor sheets in GrFs with varied proportion of 8-layer sheets. It is found that the stretching, bending and van der Waals energy would increase with the proportion of 8-layer sheets in GrFs. Furthermore, we examine the distribution of local von-Mises stress and the local stretching/shearing/bending strain of bi-modal GrFs. We find that a large number of high-stress regions non-uniformly distribute in 8-layer sheets and that only a small portion of them emerges in 1-layer sheets. In contrast, a large number of high-strain regions, no matter the local stretching, shearing or bending strain of sheets, non-uniformly locate in 1-layer sheets. Moreover, the elastic recovery of GrFs can be effectively enhanced by adding more 8-layer sheets in GrFs. These results would be helpful for us to understand and further guide the design of GrFs composed of both flexible and stiff graphene sheets.

## Figures and Tables

**Figure 1 materials-14-05622-f001:**
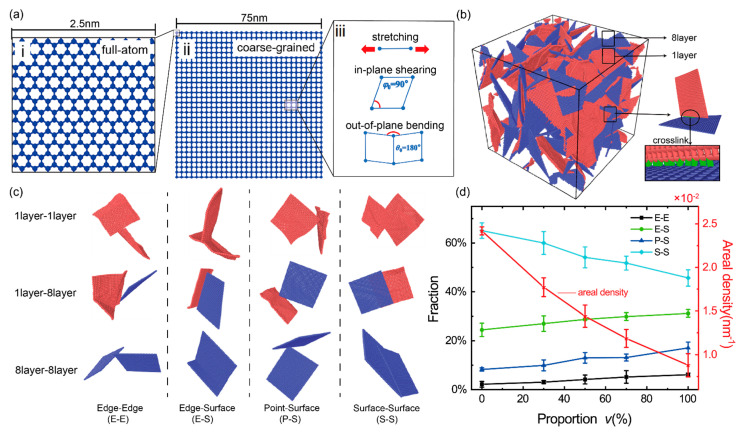
Coarse-grained model of bi-modal GrFs. (**a-i**) A square full-atomic graphene sheet with a side length of 2.5 nm; (**a-ii**) A square coarse-grained graphene sheet with a side length of 75 nm; (**a-iii**) three deformation modes of stretching, in-plane shearing and out-of-plane bending of graphene sheet; (**b**) The initial state of a well-equilibrated coarse-grained GrF consisting of the same number of 1-layer flexible (red) and 8-layer stiff (blue) coarse-grained graphene sheets; The crosslinks (green) are added between neighbor beads in different sheets. (**c**) Four typical microstructures of edge–edge (E–E), edge–surface (E–S), point–surface (P–S) and surface–surface (S–S) for two 1-layer sheets, a 1-layer sheet and a 8-layer sheet, and two 8-layer sheets; (**d**) The fraction of the four typical microstructures and the areal density (the contact area between sheets per unit volume) as an increased proportion of stiff sheets (the number of stiff sheets divided by the total number of sheets).

**Figure 2 materials-14-05622-f002:**
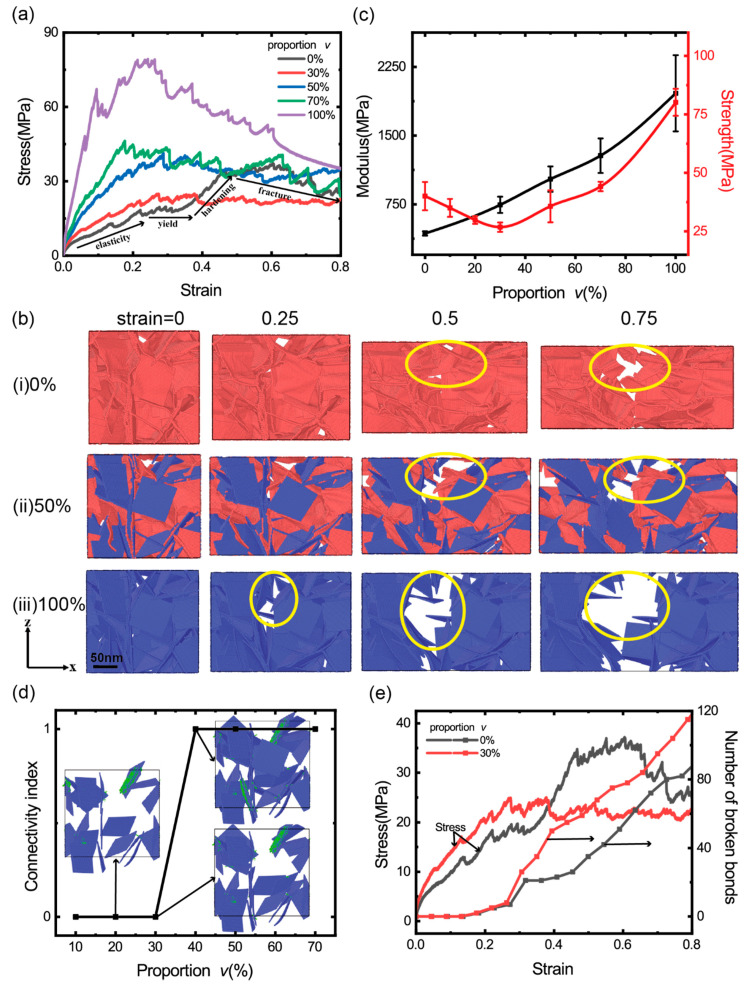
(**a**) The uniaxial tensile stress–strain curves of the GrFs with varied proportions of 8-layer stiff sheets *v* = 0, 30%, 50%, 70% and 100%; (**b**) The snapshots of the microstructural evolution of the GrFs with *v* = 0, 50% and 100% at four tensile strains of 0, 0.25, 0.5 and 0.75, respectively; (**c**) The modulus and strength of the GrFs as a function of the proportion of stiff sheets. (**d**) The connectivity of the GrFs with varied proportions of stiff sheets (The connectivity of the system is set to be 1 if the connected stiff sheets can run through the entire system in the loading direction; otherwise, it is 0); (**e**) The stress and the number of broken bonds in the GrFs with *v* = 0 and 30% as a function of tensile strain.

**Figure 3 materials-14-05622-f003:**
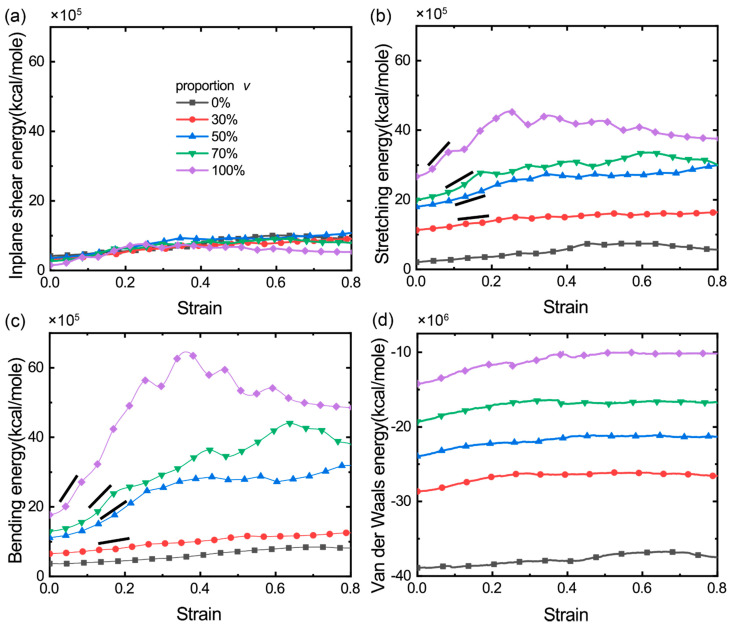
The distribution and evolution of three deformation energies, of stretching (**a**), in-plane shearing (**b**) and out-of-plane bending (**c**) of sheets, and the van der Waals potential energy (**d**) between neighboring sheets.

**Figure 4 materials-14-05622-f004:**
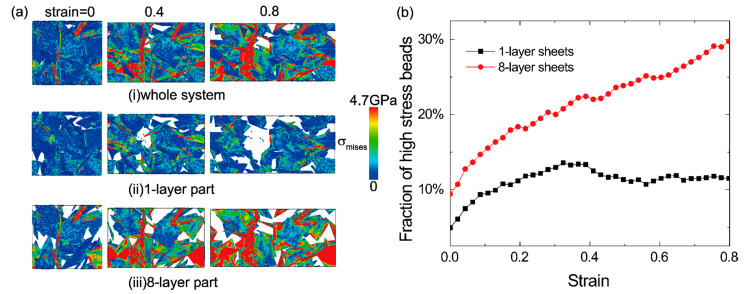
(**a-i**) The distribution of the local von-Mises stress of the whole GrF system with *v* = 50%; for a direct comparison of the stress distribution in 1-layer sheets and 8-layer sheets, the two parts are displayed separately in (**a-ii**) and (**a-iii**); (**b**) The comparison of the ratio of high stress beads in 1-layer sheets and 8-layer sheets in the GrF with *v* = 50%.

**Figure 5 materials-14-05622-f005:**
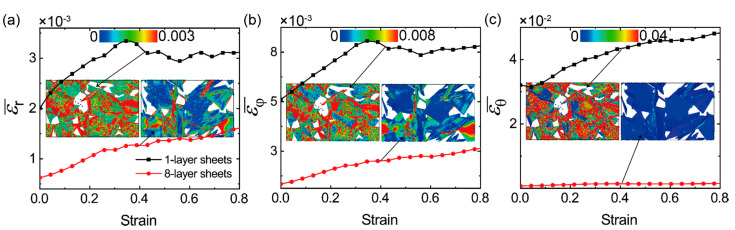
A comparison of the relative variation of (**a**) bond length, (**b**) in-plane shearing angle and (**c**) out-of-plane bending angle in 1-layer flexible sheets and 8-layer stiff sheets in the GrF with *v* = 50%, respectively. Inset: the distribution of the corresponding deformation in flexible sheets and stiff sheets at the tensile strain of 0.4.

**Figure 6 materials-14-05622-f006:**
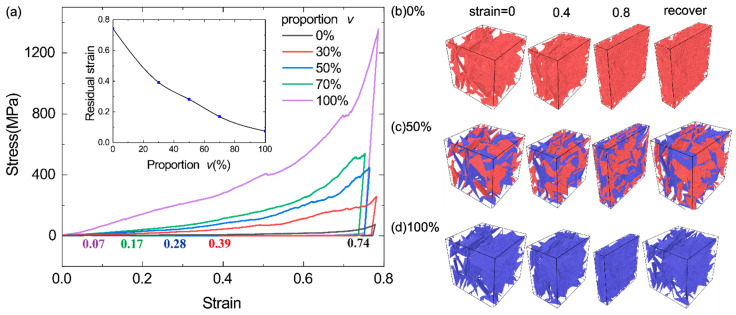
The effect of the proportion of 8-layer sheets on the elastic recovery of GrFs. (**a**) The stress-strain curves of GrFs with *v* = 0, 30%, 50%, 70% and 100% in the compression–uncompression process. Inset: the residual strain as a function of *v*; (**b**–**d**) Four typical snapshots of the GrFs with *v* = 0, 50% and 100% at the compressive strain of 0, 0.4 and 0.8 and after recovery.

**Table 1 materials-14-05622-t001:** The parameters of the main force field [44].

	No. of Graphene Layers		
Parameter	1	8	Units
$k_{B}$	235	1860	kcal mol^−1^ Å^−2^
$k_{C}$	1860	1860	kcal mol^−1^ Å^−2^
$r_{0}$	25	25	Å
$k_{\varphi }$	8435	67,480	kcal mol^−1^ rad^−2^
$\varphi_{0}$	90°	90°	——
$k_{\theta }$	72.45	466,543.5	kcal mol^−1^ rad^−2^
$\theta_{0}$	180°	180°	——
ε	473	473	kcal mol^−1^

## Data Availability

Data sharing is not applicable.

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
