# Peer review of "Mechanical Properties and Deformation Mechanisms of Graphene Foams with Bi-Modal Sheet Thickness by Coarse-Grained Molecular Dynamics Simulations"

_materials, 2021, doi:10.3390/ma14195622_

Round 1
Reviewer 1 Report
In the manuscript: “Mechanical properties and Deformation Mechanisms of Graphene Foams with Bi-modal Sheet Thickness by Coarse Grained Molecular Dynamics Simulations” by Shenggui Liu, Mindong Lyu and Chao Wang, a coarse-grained bi-modal graphene foam (GrF) model was proposed, where a mixture of 1-layer flexible and 8-layer stiff sheets were discussed their mechanical properties and deformation mechanisms using the mesoscopic model of graphene sheets.
Proposed subject is interesting to wide scientific audience, due to attentions that GrF attracted and its potential applications in energy storing and stretchable devices.
The authors presented coarse-grained molecular dynamics simulation to study and investigated the mechanical properties and deformation mechanisms of GrF with different layer number (1-8).
Although results seem convincing, the study would be improved if authors could prove experimental data for at least one of the mechanical properties.
Gramma issues: space after comma and dots
Reviewer 2 Report
The paper is devoted to theoretical analysis of mechanical properties and deformation mechanisms of graphene foam-like structure. The foam is composed of graphene sheets comprising 1 – 8 graphene layers. I fully agree that the influence of layered structure (the fact that in the graphene foam the number of layers could be substantially different from point to point) is very important for prediction of its physical and mechanical properties.
The results of molecular dynamics simulation are very interesting and important for practical applications. However, without the comparison with experimental results (authors mentioned that there are plenty of them) it is difficult to judge how good these theoretical estimations allow to describe the real graphene foam and its mechanical properties.
I would recommend adding some table or a few paragraphs with the comparison of modeling results and experimental data published by other authors.
I also do not understand why the curve in Fig.4b related to 1-layer sheet and is not monotonous? What is the physical reason for that?
Reviewer 3 Report
Materials-1371739
In this paper, the authors propose a course-grained bi-modal Gr-F model composed of a mixture of 1-layer flexible and 8-layer stiff sheets for study of the mechanical properties and deformation mechanism based on the mesoscopic model of graphene sheets.
The study is complete and rigorous, and my opinion can be published in Materials, but is necessary that the authors review and complete the conclusions
Author Response
Referee:
The study is complete and rigorous, and my opinion can be published in Materials, but is necessary that the authors review and complete the conclusion.
Our response:
We thank the reviewer for your positive comments. We have earnestly reviewed and industriously completed the conclusion after the suggestions.
Round 2
Reviewer 2 Report
I am satisfied with the revised version of the manuscript and think it can be published as it is.